# 4-Octyl itaconate inhibits aerobic glycolysis by targeting GAPDH to exert anti-inflammatory effects

Shan-Ting Liao[1,3], Chao Han[1,3], Ding-Qiao Xu[1], Xiao-Wei Fu[1], Jun-Song Wang [2]* & Ling-Yi Kong [1]*

Activated macrophages switch from oxidative phosphorylation to aerobic glycolysis, similar to the Warburg effect, presenting a potential therapeutic target in inflammatory disease. The endogenous metabolite itaconate has been reported to regulate macrophage function, but its precise mechanism is not clear. Here, we show that 4-octyl itaconate (4-OI, a cell-permeable itaconate derivative) directly alkylates cysteine residue 22 on the glycolytic enzyme GAPDH and decreases its enzyme activity. Glycolytic flux analysis by $U^{13}C$ glucose tracing provides evidence that 4-OI blocks glycolytic flux at GAPDH. 4-OI thereby downregulates aerobic glycolysis in activated macrophages, which is required for its anti-inflammatory effects. The anti-inflammatory effects of 4-OI are replicated by heptelidic acid, 2-DG and reversed by increasing wild-type (but not C22A mutant) GAPDH expression. 4-OI protects against lipopolysaccharide-induced lethality in vivo and inhibits cytokine release. These findings show that 4-OI has anti-inflammatory effects by targeting GAPDH to decrease aerobic glycolysis in macrophages.

[1] Jiangsu Key Laboratory of Bioactive Natural Product Research and State Key Laboratory of Natural Medicines, School of Traditional Chinese Pharmacy, China Pharmaceutical University, 24 Tong Jia Xiang, 210009 Nanjing, China. [2] Center for Molecular Metabolism, Nanjing University of Science and Technology, 200 Xiao Ling Wei, 210014 Nanjing, China. [3]These authors contributed equally: Shan-Ting Liao, Chao Han. *email: wang.junsong@gmail.com; cpu_lykong@126.com

Macrophages play a crucial role in innate immunity and contribute to host defence against pathogens[1–3]. An important feature of macrophages is their remarkable plasticity and ability to undergo rapid changes in morphology and status in response to their microenvironment, tailored to their functional requirements[4–7]. Pro-inflammatory stimuli lead to a Warburg-like upregulation of glycolysis in macrophages, similar to observations in tumours[3,8,9]. The switch from oxidative phosphorylation to aerobic glycolysis for energy production is important for the balance between the inflammatory and regulatory immune phenotypes of macrophages.

Glycolytic metabolism promotes the survival, differentiation and effector functions of activated macrophages[10]. Recent studies have reported that the distinct metabolic profile of macrophages controls their activation state and function[11,12]. Itaconate was first discovered to be synthesized and secreted by the fungal organism *Aspergillus terreus*[13]. It is usually used as a raw material for the chemical synthesis of polymers[14]. More recently, itaconate has been shown to be generated in lipopolysaccharide (LPS)-activated macrophages and to be produced by the mitochondria-associated enzyme immunoresponsive gene 1 (IRG1)[15,16]. Itaconate is reported to exert antibacterial effects by inhibiting isocitrate lyase, which is a bacterial glyoxylate shunt enzyme[17]. It was also reported that itaconate has a notable anti-inflammatory effect in activated macrophages[16,18,19]. Therefore, the anti-inflammatory mechanism of itaconate needs further study.

Itaconate, as a carboxylic acid, is highly polar and is not easily permeable to cell membranes, so it is not suitable for a mechanism study. To overcome the limitations of itaconate, 4-octyl itaconate (4-OI), a cell-permeable itaconate derivative, was synthesized. Itaconate and 4-OI had similar thiol reactivity, making 4-OI a suitable itaconate surrogate to study its biological function[20]. 4-OI is reported to alkylate cysteine residues on kelch-like ECH-associated protein 1 (KEAP1) and then activate nuclear factor (erythroid derived 2)-related factor 2 (Nrf2) to exert antioxidant and anti-inflammatory effects[20]. With electrophilic α, β-unsaturated moieties similar to those of 4-OI, dimethyl fumarate also exerts anti-inflammatory and antioxidant effects through posttranslational modification of Nrf2[21–23]. Endogenous fumarate also succinates the glycolytic enzyme glyceraldehyde 3-phosphate dehydrogenase (GAPDH) and inhibits its enzyme activity and thus decreases inflammation[24]. In this study, we found that 4-OI alkylated cysteine residues of GAPDH and inhibited its enzyme activity, activating the anti-inflammatory programme.

## Results

**4-OI targets cysteine 22 (Cys 22) in GAPDH.** Liquid chromatography–tandem mass spectrometry (LC-MS/MS) metabolomics was used to screen differential metabolites in RAW264.7 macrophages after LPS stimulation compared with untreated RAW264.7 macrophages. LPS strongly increased the levels of itaconate (Fig. 1a, b, Supplementary Fig. 1). Itaconate is made by diverting aconitate away from the tricarboxylic acid (TCA) cycle in activated macrophages. Itaconate or its derivatives can modify or regulate multiple proteins, including KEAP1 and ATF3, exerting their roles in inflammation[18,20]. The main reason that macrophages show this response currently appears to be an anti-inflammatory action, with itaconate linking cell metabolism, the oxidative and electrophilic stress responses and immune responses[25]. Whether itaconate exerts anti-inflammatory effects through other mechanisms remains unclear.

4-OI, a suitable cell-permeable itaconate surrogate, was used to replace itaconate. 4-OI has an electrophilic α, β-unsaturated

moieties that may alkylate the thiol in cysteine residues of proteins via the Michael addition. 4-OI could be hydrolysed to itaconate in LPS-stimulated mouse macrophages (Supplementary Fig. 2). An attractive candidate protein for cysteine residue alkylation is GAPDH, a rate-limiting enzyme in only cancer and activated immune cells, in which aerobic glycolysis is upregulated in the setting of Warburg physiology[26,27]. To verify whether GAPDH was alkylated by 4-OI, LC-MS/MS was performed. In RAW264.7 macrophages treated with 4-OI, Cys 22 of GAPDH was alkylated by 4-OI (Fig. 1c and Supplementary Table 1). Alkylation of GAPDH Cys 22 was also identified in LPS-stimulated RAW264.7 macrophages treated with itaconate (Fig. 1d and Supplementary Table 2). For verification, recombinant mouse GAPDH was used; 4-OI alkylated Cys 22 (Fig. 1e and Supplementary Table 3) and Cys 245 (no functional influence) (Supplementary Fig. 3a, b).

**GAPDH inhibition by 4-OI ameliorates aerobic glycolysis.** In activated macrophages, GAPDH is a rate-limiting enzyme that regulates the rate of aerobic glycolysis[27]. To investigate whether 4-OI-induced alkylation of Cys 22 of GAPDH influences its enzymatic activity, GAPDH activity in RAW264.7 cells was detected. GAPDH activity was inhibited in LPS-induced macrophages after 4-OI treatment for 24 h (Fig. 2a). Enzyme activity was further measured for recombinant mouse GAPDH, showing time- and dose-dependent decreases by 4-OI treatment (Fig. 2b). These results indicated that the 4-OI-induced posttranscriptional modification of Cys 22 of GAPDH inhibits its enzymatic activity, thereby regulating aerobic glycolysis.

Lactate, the end-product of glycolysis, was used as a marker for glycolysis[28], which is decreased in a dose-dependent manner by 4-OI treatment in RAW264.7 macrophages and bone marrow-derived macrophages (BMDMs) stimulated with LPS for 24 h (Fig. 2c, d). Furthermore, 4-OI had no effect on cell viability (Supplementary Fig. 3c, d). In activated macrophages, pyruvate is blocked from entering into the TCA cycle, which favours lactate production[29]. The increased extracellular acidification rate (ECAR) (Fig. 2e, f) and decreased oxygen consumption rate (OCR) (Fig. 2g, h) were significantly relieved by 4-OI treatment in the two macrophage groups, showing that 4-OI significantly inhibited the switch from oxidative phosphorylation to glycolysis in LPS-activated macrophages in a concentration-dependent manner. *Irg1* is the gene coding for an enzyme-producing itaconic acid by the decarboxylation of *cis*-aconitate[30], so we further detected GAPDH activity and glycolysis in wild-type (WT) and *Irg1*$^{-/-}$ BMDMs. GAPDH activity was significantly enhanced in *Irg1*$^{-/-}$ BMDMs as compared with WT BMDMs after LPS stimulation for 24 h (Supplementary Fig. 4a), which together with the significant increased levels of lactate and ECAR in LPS-induced *Irg1*$^{-/-}$ BMDMs (Supplementary Fig. 4b, c), demonstrating an obviously augmented glycolysis in *Irg1*$^{-/-}$ BMDMs. As a consequence, the level of IL-1β was significantly increased in LPS-induced *Irg1*$^{-/-}$ BMDMs (Supplementary Fig. 4d). These results provided convincible evidences that the inhibition of endogenous itaconate production increased GAPDH activity and glycolysis and promote inflammation.

Next, we traced the metabolism of U$^{13}$C-glucose and found that 4-OI treatment increased the level of dihydroxyacetone phosphate (the upstream metabolite of GAPDH) and reduced the level of lactate (the downstream metabolite of GAPDH) in LPS-stimulated BMDMs. These results showed that 4-OI generated a blockade in glycolytic flux at GAPDH (Fig. 2i), providing evidence that inhibition of GAPDH activity mediated the downregulation of glycolysis by 4-OI.

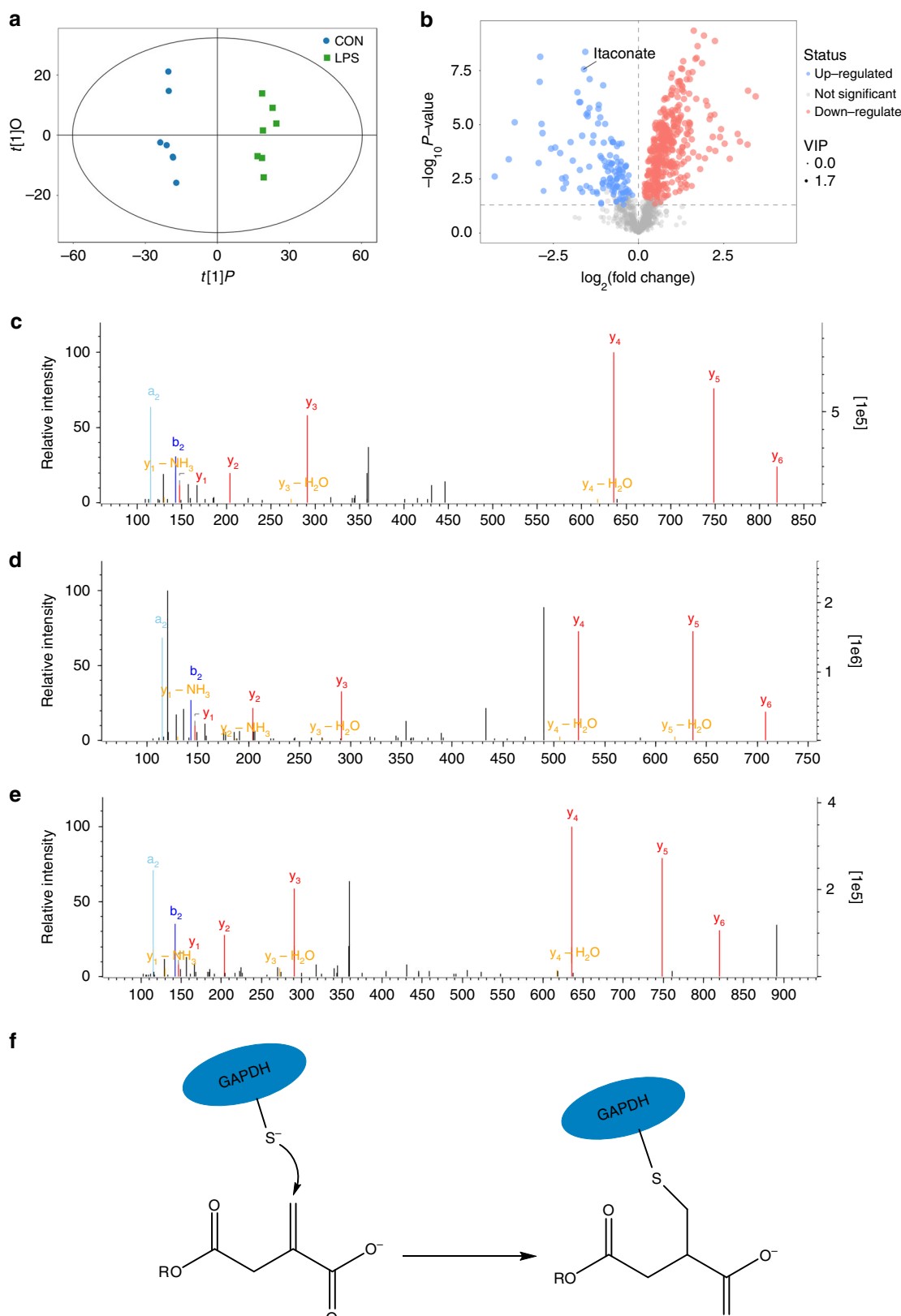

**GAPDH inhibition by 4-OI alleviates inflammation**. We next assessed the subsequent effects of the inhibition of GAPDH activity and aerobic glycolysis by 4-OI treatment. 4-OI significantly inhibited LPS-induced increases in the protein (Fig. 3a–d) and mRNA levels (Supplementary Fig. 5) of interleukin (IL)-1β and inducible nitric oxide synthase (iNOS) in both RAW264.7 macrophages and BMDMs, showing that 4-OI prevented macrophage activation. To determine whether aerobic glycolysis mediated macrophage activation, 2-deoxy-D-glucose (2-DG), an extensively used competitive inhibitor for the first hexokinase of the glycolytic pathway, was used. 2-DG switch to 2-DG-P by phosphorylation of hexokinase, which cannot be further

**Fig. 1** 4-OI alkylates cysteine 22 of GAPDH. **a** The orthogonal projection to latent structures discriminant analysis (OPLS-DA) score plots compared control (CON) and LPS-stimulated (LPS) RAW264.7 macrophages (24 h). Results are from seven independent experiments. **b** Metabolite levels in CON versus LPS-induced RAW264.7 macrophages. Blue and red dots represent metabolites significantly downregulated and upregulated after LPS stimulation, respectively. **c** Representative LC-MS/MS spectra showing covalent modification of the cysteine 22-containing GAPDH peptide by 4-OI (+242.15 Da) after 4-OI treatment for 4 h in GAPDH immunoprecipitated from RAW264.7 macrophages. **d** Representative LC-MS/MS spectra showing covalent modification of the cysteine 22-containing GAPDH peptide by itaconate (+130.02 Da) after LPS stimulation for 6 h in GAPDH immunoprecipitated from RAW264.7 macrophages. **e** Representative LC-MS/MS spectra showing covalent modification of the cysteine 22-containing GAPDH peptide by 4-OI (+242.15 Da) in recombinant mouse GAPDH treated with 4-OI (500 μM) for 4 h at 37 °C. Da daltons. **f** Schematic diagram of GAPDH thiol group was covalent modified by 4-OI

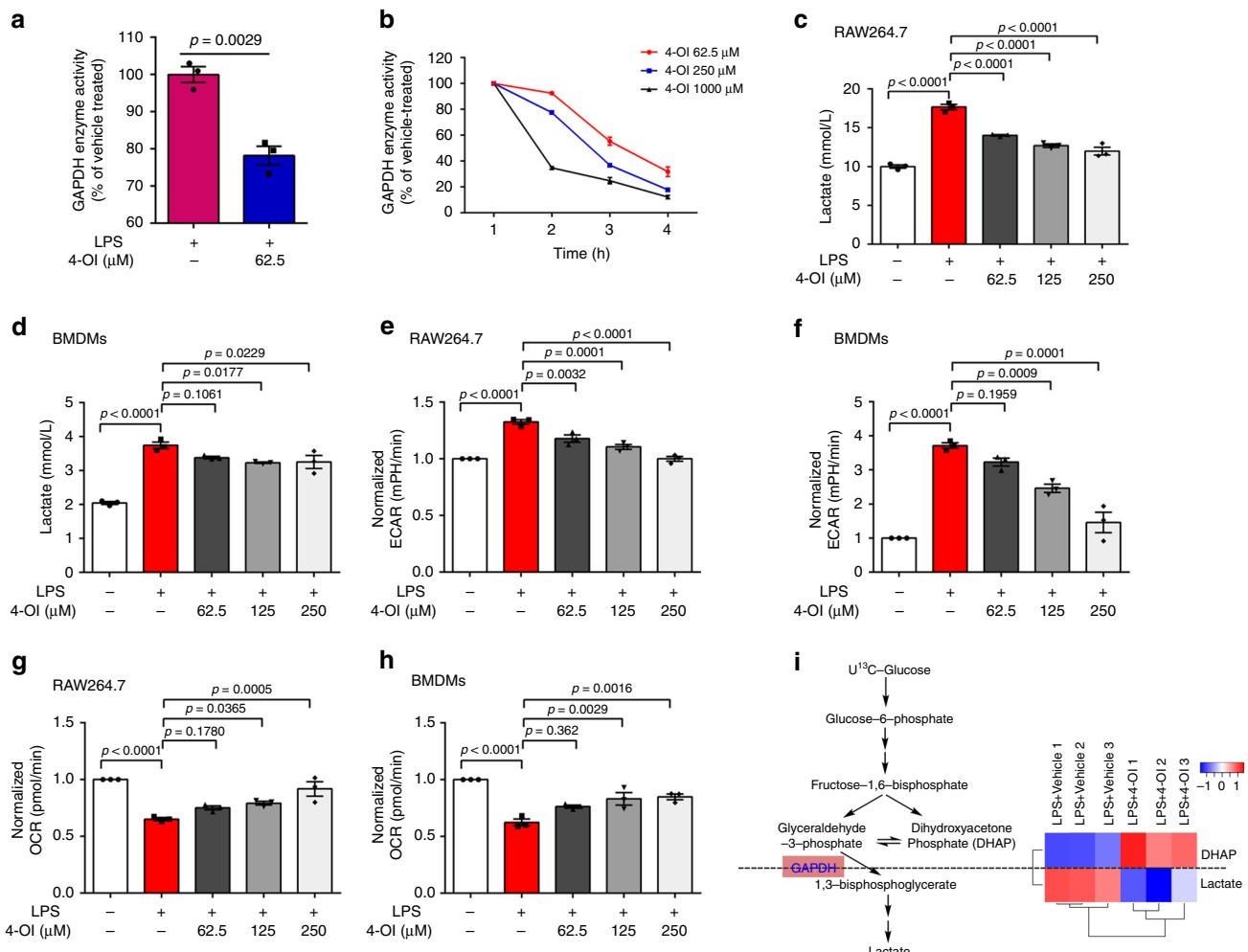

**Fig. 2** 4-OI inhibits GAPDH activity and glycolysis. **a** RAW264.7 macrophages were treated with 4-OI (62.5 μM) for 3 h and then subjected to LPS stimulation for 24 h. GAPDH enzyme activity assays of cell lysates were performed. **b** 4-OI dose- and time-dependent inhibition of GAPDH enzyme activity. Recombinant mouse GAPDH was incubated with the indicated drug concentrations. Aliquots were removed at various time points, followed by an enzyme activity assay. **c–h** RAW264.7 macrophages and BMDMs were treated with 4-OI for 3 h, followed by LPS stimulation for 24 h. **c, d** Lactate levels of RAW264.7 macrophages (**c**) and BMDMs (**d**) were determined by a lactate assay. **e, f** ECAR of RAW264.7 macrophages (**e**) and (**f**) BMDMs were measured by an ECAR assay as described in "Methods." **g, h** OCR of RAW264.7 macrophages (**g**) and BMDMs (**h**) were assayed by an OCR assay as described in "Methods." **i** RAW264.7 macrophages were treated with 125 μM 4-OI or vehicle and stimulated with 1 μg/mL LPS, in triplicate. Cells were then added with 12 mM U$^{13}$C-glucose, and $^{13}$C-glucose labelling of glycolytic intermediates was measured by GC-MS. Colour key in heat map indicates the metabolite expression value: red represents the significant increases and blue represents the significant decreases. Heat map indicated blockade of glycolytic flux at GAPDH. All data shown are summarized from three independent experiments. Values represent the mean ± SEM at each time point. $p$ Values were calculated using two-tailed Student's $t$ test or one-way ANOVA with Sidak's correction for multiple comparisons. Source data are provided as a Source Data file

metabolized by phosphoglucose isomerase. Blockade of glycolysis by 2-DG also decreased LPS-induced IL-1β production, thereby inhibiting macrophage activation (Fig. 3e).

Aerobic glycolysis uses glucose to produce energy, so the concentration of glucose determines the glycolysis rate[23].

Therefore, we investigated whether the influence of glucose concentrations mediated the anti-inflammatory effect of 4-OI. We examined the release of IL-1β in BMDMs under low (0.5 mM) or high (25 mM) glucose concentrations (Fig. 3f). The decreases in IL-1β caused by 4-OI treatment in LPS-induced

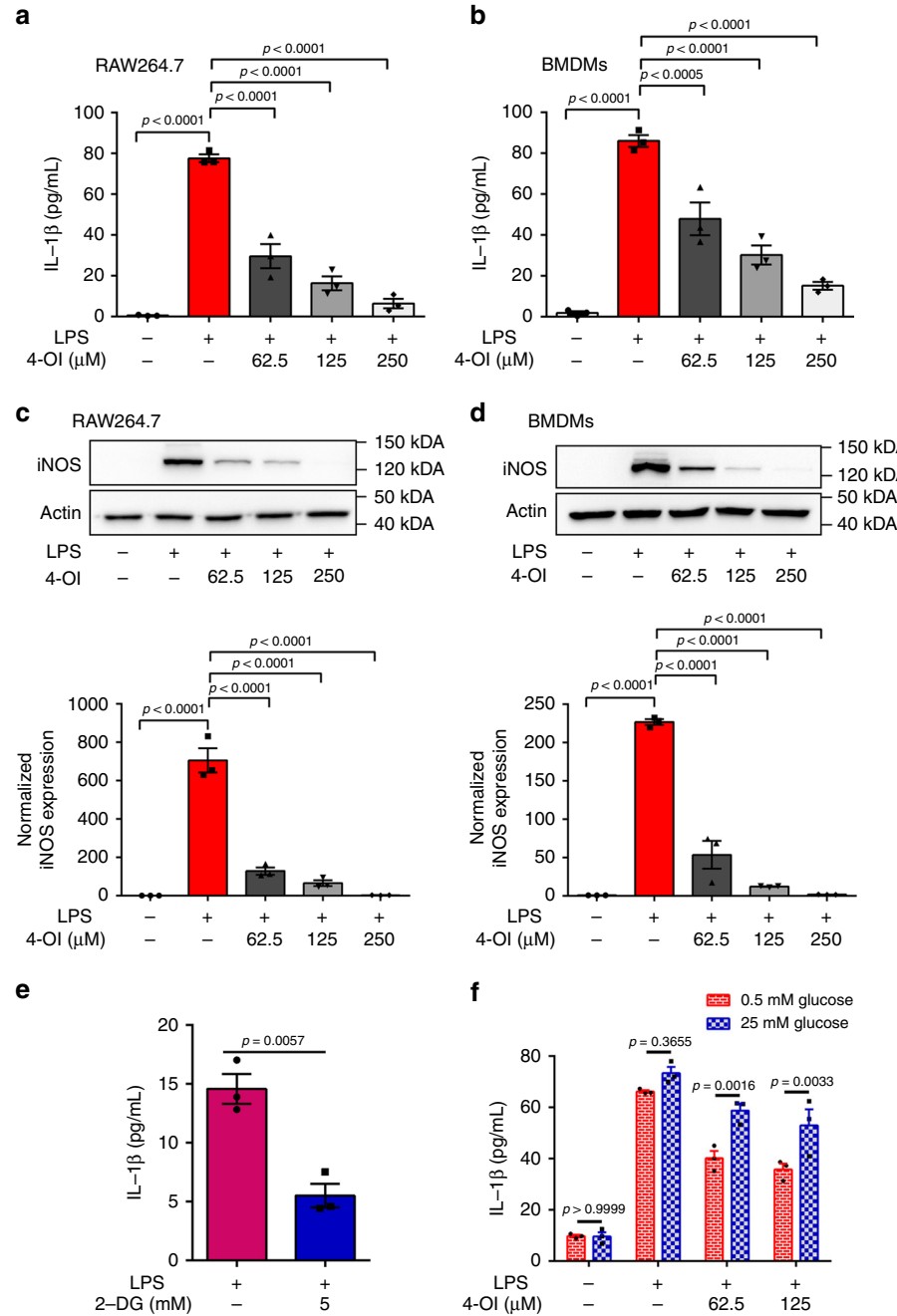

**Fig. 3** 4-OI alleviates inflammation by inhibiting GAPDH activity. **a**, **b** RAW264.7 macrophages (**a**) and BMDMs (**b**) were treated with vehicle or 4-OI at the indicated concentrations. After 3 h, RAW264.7 macrophages or BMDMs were stimulated with LPS (1 μg/mL or 100 ng/mL) for 24 h, and IL-1β in the supernatants was measured by ELISA. **c**, **d** LPS-induced iNOS protein expression and its relative quantification after 24 h in RAW264.7 macrophages (**c**) and BMDMs (**d**) pretreated with or without 4-OI for 3 h. Data were corrected based on the actin loading control. **e** Treating LPS-stimulated RAW264.7 macrophages with 5 mM 2-DG, a glycolysis inhibitor, replicated the effect of 4-OI on IL-1β production. **f** IL-1β secretion was measured by ELISA in BMDMs that were treated with LPS ± DMF for 24 h in either limiting (0.5 mM) or saturating (25 mM) concentrations of glucose. Data shown are representative of three independent experiments. *p* Values were calculated using one-way ANOVA with Sidak's correction or two-way ANOVA with Turkey's correction for multiple comparisons tests. All data show mean ± SEM. Source data are provided as a Source Data file

BMDMs could be attenuated by high concentrations of glucose, suggesting that the anti-inflammatory effect of 4-OI can be overcome by driving glycolysis with saturating concentrations of glucose. These results suggest that 4-OI may exert anti-inflammatory effects by mediating glycolysis.

**4-OI inhibits inflammation by modifying C22 of GAPDH.** GAPDH specifically catalyses oxidative phosphorylation of NAD[+]

and glyceraldehyde-3-phosphate to 1,3-biphosphoglycerate and NADH most strongly correlates with aerobic glycolysis[27,31]. Heptelidic acid (a GAPDH inhibitor) replicated the effects of 4-OI on IL-1β production and iNOS expression (Fig. 4a–c). Heptilidic acid was reported with significant inhibition on the translation of tumour necrosis factor (TNF)-α but not its transcription[32]. Our results showed that the performance of 4-OI is indeed similar to that of heptilidic acid: inhibition of TNF-α translation and has no effect on its mRNA levels (Fig. 4d, e). Furthermore, 4-OI, 2-DG

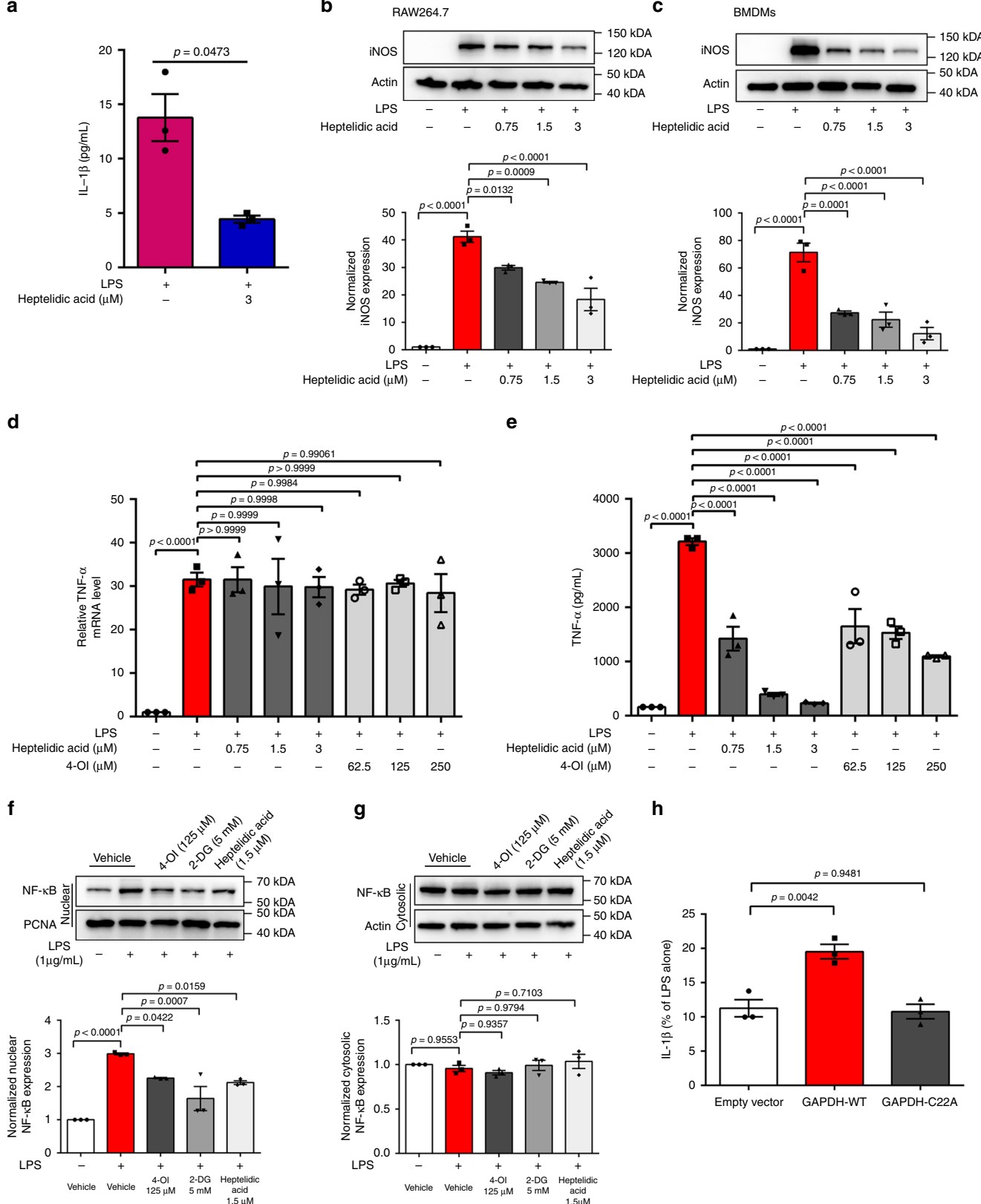

and heptelidic acid both inhibited nuclear translocation of nuclear factor-κB (NF-κB) (Fig. 4f, g), resulting in decreased inflammation. To further confirm the important role of Cys-22 in GAPDH, we overexpressed empty vector, WT GAPDH (GAPDH-WT) and Cys-22 mutant GAPDH (GAPDH-C22A) in RAW264.7

macrophages. The inhibitory effect of 4-OI on IL-1β secretion (Fig. 4h) and mRNA levels (Supplementary Fig. 6) was rescued by overexpression of GAPDH-WT but not GAPDH-C22A. These data showed that the anti-inflammatory effects of 4-OI were indeed due to its alkylation of the C22 residue of GAPDH.

**Fig. 4** Covalently modifying C22 of GAPDH mediates the anti-inflammatory effects of 4-OI. **a** LPS-stimulated RAW264.7 macrophages were treated with 3 μM heptelidic acid, a GAPDH inhibitor, which also decreased IL-1β release. **b**, **c** Treatment with heptelidic acid replicated the effects of 4-OI on iNOS expression in LPS-stimulated RAW264.7 macrophages (**b**) and BMDMs (**c**). **d**, **e** TNF-α mRNA expression and protein release were measured in 1 μg/mL LPS-treated (6 h) RAW264.7 cells after treatment with heptelidic acid or 4-OI. **f**, **g** LPS-induced NF-κB (24 h) expression in the nucleus (**f**) or cytosol (**g**) after treatment with vehicle or 4-OI as indicated. **h** Wild-type GAPDH (GAPDH-WT) or the Cys-22 mutant (GAPDH-C22A) was overexpressed in RAW264.7 macrophages for 24 h and then treated with LPS and 4-OI for 24 h, followed by measurement of IL-1β secretion by ELISA. Representative results are shown from three independent experiments. Data represent the mean ± SEM. *p* Values were determined by two-tailed Student's *t* test or one-way ANOVA with Sidak's correction for multiple comparisons test. Source data are provided as a Source Data file

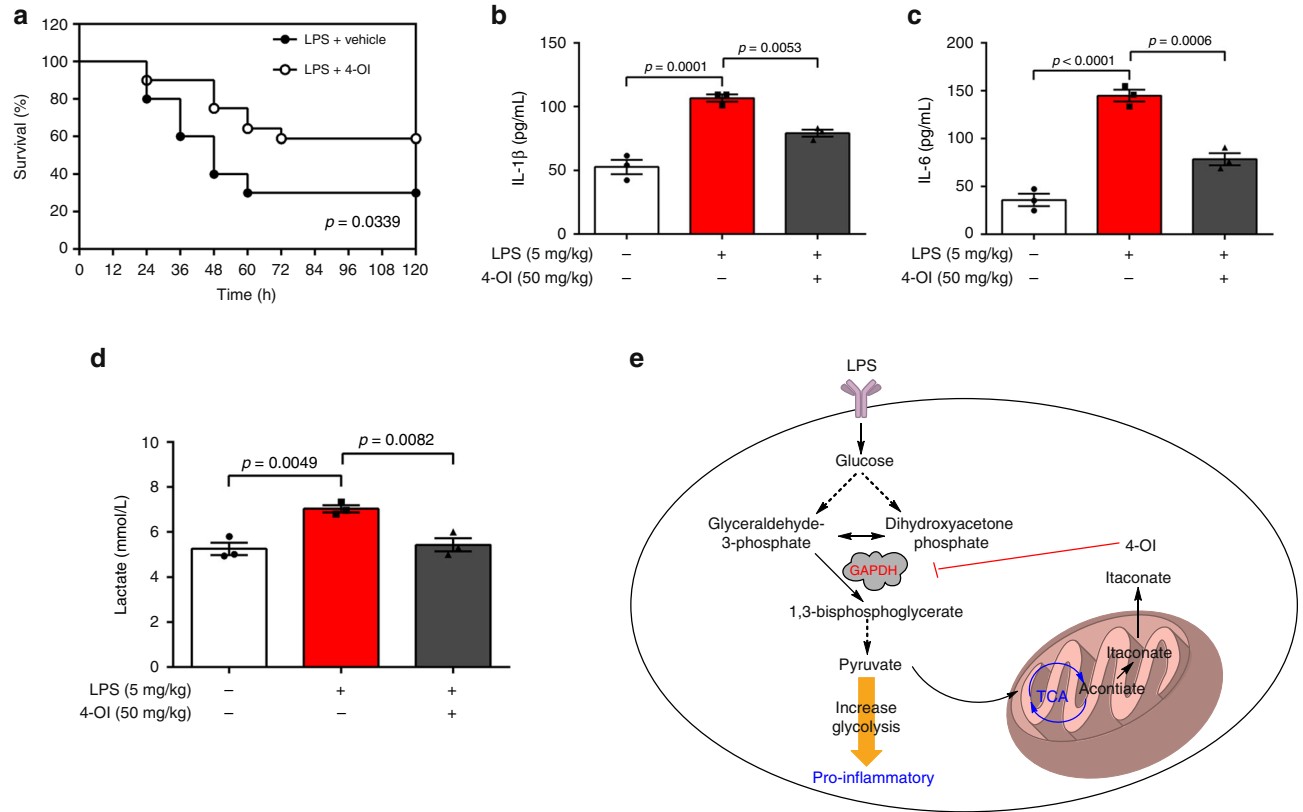

**Fig. 5** 4-OI protects against LPS lethality. **a** Mice were injected with a single dose of 4-OI (50 mg/kg), followed 2 h later by an intraperitoneal injection of LPS (5 mg/kg), and the mice were then re-treated with 4-OI 24 and 48 h later (*n* = 20 mice/group). Log-rank (Mantel–Cox) test was used to compare the differences in survival rates between groups. **b–d** In parallel experiments, the serum levels of IL-1β (**b**) and IL-6 (**c**) at 2 h and lactate (**d**) at 24 h after intraperitoneal injection of LPS (5 mg/kg) were measured. Results are from three independent experiments. Data values are the mean ± SEM, *p* values were calculated by one-way ANOVA with Sidak's correction for multiple comparisons test. Source data are provided as a Source Data file. **e** Proposed model of the anti-inflammatory role of 4-OI, which may explain the physiologic negative feedback function of itaconate

**4-OI suppresses the pro-inflammatory response to LPS in vivo.** Given the capacity of 4-OI to attenuate LPS-stimulated IL-1β production in vitro, we asked whether 4-OI protects mice against lethal endotoxaemia by decreasing mortality and inhibiting cytokine release. C57BL/6J mice were administered 4-OI (50 mg/kg) or vehicle 2 h before intraperitoneal injection of LPS (5 mg/kg). 4-OI treatment significantly prolonged the survival rate (Fig. 5a) and simultaneously decreased the serum levels of IL-1β (Fig. 5b), IL-6 (Fig. 5c) and lactate (Fig. 5d) in mice induced by LPS. These results demonstrated that 4-OI protects mice against experimental lethal endotoxaemia partly by inhibiting cytokine release and lactate production.

## Discussion
Cellular metabolic adaptation to immune responses, known as immunometabolism, plays an important role in regulating the immune function of cells[12,15,19]. In recent years, substantial and growing progress has been made in the field of immunometabolism[33,34], providing an in-depth understanding of the role of intracellular metabolites (known as immunometabolites) in the complex regulation of immune cells. Macrophages, a main component of innate immunity, play a vital role in host defence and homeostasis[35]. Macrophages are the first line of defence for the immune system and contribute to defence against infection by producing pro-inflammatory factors (such as IL-1β)[36]. In response to external stimuli, the levels of many metabolites in macrophages are changed. Metabolites are substrates and products of biochemical reactions, reflecting and participating in enzyme activity. It is thus important to study the roles of these metabolites in the function of macrophages.

The Warburg effect, initially found in various tumour cells, is also observed in activated macrophages[8,37]. There are two major types of macrophages: the classical M1 phenotype, which can be activated by LPS or interferon-γ to trigger pro-inflammatory responses, and the alternate M2 phenotype, which is activated by IL-4, IL-10 or IL-13 to execute anti-inflammatory effects[38,39]. LPS

induces the metabolic switch from oxidative phosphorylation to aerobic glycolysis in M1 macrophages even in an oxygen-rich environment, which contributes to the transcriptional expression and release of early pro-inflammatory cytokines, e.g. IL-1β[40,41]. M1 macrophages are characterized by high proliferation and rapid responses to different immune stimuli[34]. Aerobic glycolysis is adopted by M1 macrophages to meet their high demand for quick energy and an ample supply of biosynthetic raw material, satisfying the needs of proliferation and biosynthesis[42,43]. Itaconate, as an immunometabolite, has been reported to have a significant anti-inflammatory effect[18,20,30]. However, the molecular mechanisms underlying this effect are not fully understood. In this study, we found that 4-OI significantly inhibited aerobic glycolysis and IL-1β production induced by LPS. Intriguingly, the anti-inflammatory effects of 4-OI were attenuated by high concentrations of glucose-induced glycolysis enhancement, linking the anti-inflammatory effects of 4-OI to its inhibition of glycolysis.

GAPDH catalyses the conversion of glyceraldehyde 3-phosphate to D-glycerate 1,3-bisphosphate, the sixth step in the glycolytic breakdown of glucose in the cytosol of eukaryotic cells[31]. GAPDH is an irreversible metabolic switch in glycolysis, which is desperately needed only in rapidly proliferating cells[27]. Significant upregulation of glycolysis and increased activity of GAPDH are characteristics of activated macrophages[23]. Therefore, GAPDH might be an attractive anti-inflammatory target: inhibition of GAPDH activity could suppress aerobic glycolysis and, as a consequence, inhibit inflammation. Many metabolites containing an α, β-unsaturated carboxylic acid could alkylate cysteine residues of proteins, such as fumarate on KEAP1, and GAPDH and 4-OI on KEAP1, by the Michael addition. Therefore, the anti-inflammatory effect of 4-OI might also arise from its alkylation of cysteine residues of GAPDH. 4-OI could covalently modify KEAP1 by dicarboxypropylation[20]. This modification increased nucleus Nrf2 level and facilitated the expression of downstream target genes with anti-inflammatory and antioxidant capacities. KEAP1 normally forms complex with Nrf2 and promotes its degradation. Alkylation of crucial KEAP1 cysteine residue by 4-OI leads to the accumulation of newly synthesized Nrf2, which migrate to the nucleus and activate a transcriptional antioxidant and anti-inflammatory programme. Nrf2 activation is thus essential for the anti-inflammatory effect of 4-OI. In our study, 4-OI could modify the Cys 22 residue of GAPDH by similar dicarboxypropylation. The decrease in IL-1β release induced by 4-OI treatment was successfully and significantly attenuated by overexpression of WT GAPDH but not C22A GAPDH in RAW264.7 macrophages, which demonstrated the essential role of Cys 22 in GAPDH function. The anti-inflammatory effect of 4-OI is associated with the inhibited glycolysis, which provides prerequisite energy and biosynthetic raw material for M1 macrophages, helping their proliferation and biosynthesis.

Appropriate inflammatory responses promote the activation of the innate immune system against infections; however, excessive inflammation is harmful and even lethal[44,45]. Sepsis is a severe systemic inflammatory reaction resulting from harmful or lethal host responses to infections[46]. Continuously excessive inflammation in sepsis causes cell and tissue damage, multiple organ failure, and ultimately death. Inflammatory responses are largely mediated by cytokines, which are released into the systemic circulation during infection[47]. In this study, 4-OI treatment markedly improved the survival of mice with lethal endotoxaemia. IL-1β is the principal pro-inflammatory cytokine produced in response to infectious insults, and IL-6 is one of the main indicators of patients with sepsis[48,49]. 4-OI conferred significant protection against lethal endotoxaemia in mice, partly by its

significant inhibition of the release of these pro-inflammatory cytokines. Notably, elevated lactate levels in serum have been known as a biomarker of mortality and organ failure in sepsis and other critical illnesses. Increased lactate levels also feature enhanced glycolysis, which was significantly attenuated by 4-OI in LPS-induced mice[50,51]. 4-OI-induced alkylation of the Cys 22 residue of GAPDH functionally affected GAPDH activity to inhibit glycolysis, and inhibited glycolysis hampered the activation of macrophages, with decreased production of pro-inflammatory cytokines and inhibited inflammatory responses (Fig. 5e), which is favourable for survival in severe inflammatory conditions or diseases.

Accumulating evidence suggests the important roles of immunometabolites for the functions of immune cells. We demonstrated that itaconate, as an inflammatory regulator, could directly inhibit GAPDH activity through a newly identified posttranslational modification via a chain of subsequent alterations, with anti-inflammatory effects. These results offered a novel insight into the mechanisms underlying the metabolic programming of immunometabolism by itaconate and emphasized the importance of targeting aerobic glycolysis, e.g. GAPDH, in the treatment of inflammatory diseases.

## Methods

**Reagents.** LPS (*Escherichia coli* LPS 055:B5; L6529) was purchased from Sigma (St. Louis, MO, USA). The Actin antibody (4970P), the proliferating cell nuclear antigen antibody (13110P), NF-κB p65 antibody (8242S) and normal rabbit IgG antibody (7074P2) were purchased from Cell Signaling Technology (Beverly, MA). iNOS antibody (ab178945) and heptelidic acid (ab8895) were obtained from Abcam (Cambridge, MA). 2-DG (HY-13966) and Protein A/G Magnetic Beads (HY-K0202) were purchased from MedChem Express (New Jersey, USA). Hieff® qPCR SYBR® Green Master Mix (11201ES03) was obtained from Yeason (Shanghai, China). Recombinant murine macrophage colony-stimulating factor (M-CSF; 315-02) was purchased from PeproTech (London, UK).

**Cell culture.** Murine-derived macrophage RAW 264.7 cells were purchased from the Cell Bank of the Chinese Academy of Sciences (Shanghai, China). BMDMs were isolated from the leg bones of C57BL/6J mice and differentiated in Dulbecco's modified Eagle's medium (DMEM; supplemented with 10% heat-inactivated foetal bovine serum (FBS), 1% (v/v) penicillin/streptomycin and 25 ng/mL M-CSF) for 6 days. RAW 264.7 macrophages and BMDMs were cultured in DMEM (containing 10% heat-inactivated FBS and 1% (v/v) penicillin/streptomycin) at 37 ℃, 95% humidity and 5% CO₂. Unless stated otherwise, the LPS concentration used was 1 μg/mL for RAW 264.7 cells and 100 ng/mL for BMDMs. For in vitro experiments, 4-OI, 2-DG and heptelidic acid pretreatments occurred for 3 h before LPS stimulation.

**Experimental animals and model of endotoxaemia.** C57BL/6J WT mice were purchased from Zhejiang Vital River Laboratory Animal Technology Co., Ltd. (Jiaxing, China). *Irg1*−/− mice on the C57BL/6J genetic background were bought from Jackson Laboratory. Six-to-12-week-old male *Irg1*−/− mice and age-matched male WT mice were used in our experiments. This study was approved by the Institutional Animal Care and Use Committee of China Pharmaceutical University Experimental Animal Center. Endotoxaemia was induced by intraperitoneal injection of 5 mg/kg LPS in C57BL/6J mice (male, 6–7 weeks old, 20–25 g). 4-OI dissolved in 40% cyclodextrin in phosphate-buffered saline (PBS) or vehicle was administered intraperitoneally for 2 h before stimulation with LPS (5 mg/kg) intraperitoneally. Blood was collected 2 h after LPS stimulation, placed at room temperature for 2 h and then centrifuged for 15 min at 4 ℃ and 1500 × g. Supernatants were collected and deposited at −80 ℃ until analysis. Survival was recorded for up to 5 days after the onset of endotoxaemia.

**Immunoprecipitation analysis.** Cells were lysed at 4 ℃ in RIPA buffer (Cell Signaling Technology) for 30 min. Samples containing 1 μg of total protein were pre-cleared with an anti-GAPDH antibody. The tube was rotated for 2 h at 4 ℃. Then 400 μL of wash buffer was added to the beads and pipetted gently to mix. The tube was placed into a magnetic stand to collect the beads, and the supernatant was discarded. This step was repeated four times. Protein A/G Magnetic Beads were added to the samples, which were incubated with rotation for 2 h at 4 ℃. The beads were washed three times with wash buffer, and the immune complexes were eluted from the beads. Then the immune complexes were subjected to sodium dodecyl sulfate–polyacrylamide gel electrophoresis (SDS–PAGE) and used for further analysis.

**LC-MS/MS metabolomic analysis**. The cell samples were supplemented with 1 mL of extraction liquid [methanol:acetonitrile:water = 2:2:1 (v:v:v)] and 20 μL of internal standard. The samples were homogenized in a ball mill for 4 min at 45 Hz and then ultrasound treated for 5 min (incubated in ice water). The samples were incubated for 1 h at −20 °C to precipitate proteins after homogenization 3 times. The extracts were centrifuged at $13,680 \times g$ for 15 min at 4 °C. The supernatant (700 μL) was transferred into EP tubes and dried in a vacuum concentrator without heating. The residues were added to 100 μL of extraction liquid [acetonitrile: water = 1:1 (v:v)]. The samples were vortexed for 30 s and sonicated for 10 min (4 °C water bath). The mixtures were subsequently centrifuged for 15 min at $13,680 \times g$ and 4 °C, and the supernatant (60 μL) was then transferred into a fresh 2 mL LC/MS glass vial for ultra-performance LC (UPLC)-quadrupole time-of-flight (Q-TOF)-MS analysis.

LC-MS/MS analyses were performed using a UPLC system (1290, Agilent Technologies) with an UPLC BEH Amide column (1.7 μm, $2.1 \times 100$ mm², Waters) coupled to Triple TOF 5600 (Q-TOF, AB Sciex). The mobile phase consisted of 25 mM $NH_4OAc$ and 25 mM $NH_4OH$ (pH = 9.75) in water (A) and acetonitrile (B), with the elution gradient as follows: 0 min, 95% B; 7 min, 65% B; 9 min, 40% B; 9.1 min, 95% B; and 12 min, 95% B, which was delivered at 0.5 mL/min. The injection volume was 2 μL (negative) or 3 μL (positive). The Triple TOF mass spectrometer was used for its ability to acquire MS/MS spectra on an information-dependent basis during an LC/MS experiment. In this mode, the acquisition software (Analyst TF 1.7, AB Sciex) continuously evaluates the full-scan survey MS data as it collects and triggers the acquisition of MS/MS spectra depending on preselected criteria. In each cycle, 12 precursor ions with an intensity >100 were chosen for fragmentation at a collision energy of 30 V (15 MS/MS events with a product ion accumulation time of 50 ms each). The electrospray ionization source conditions were set as follows: Ion source gas 1 as 60 Psi, Ion source gas 2 as 60 Psi, Curtain gas as 35 Psi, source temperature 650 °C, and Ion Spray Voltage Floating 5000 or −4000 V in positive or negative mode, respectively.

Raw MS data were converted to the mzXML format using Proteo Wizard and processed by the R package XCMS (version 3.2). The preprocessing results generated a data matrix that consisted of the retention time, mass-to-charge ratio ($m/z$) values and peak intensity. The R package CAMERA was used for peak annotation after XCMS data processing. An in-house MS² database was applied for metabolite identification. $X$ peaks were detected, and $X$ metabolites could be left through the interquartile range denoising method. Then the missing values of raw data were filled up by half of the minimum value. In addition, an internal standard normalization method was employed in this data analysis. The resulting three-dimensional data involving the peak number, sample name and normalized peak area were fed to the SIMCA14.1 software package (V14.1, Sartorius Stedim Data Analytics AB, Umea, Sweden) for principal component analysis (PCA) and orthogonal projections to latent structures discriminate analysis (OPLS-DA). PCA showed the distribution of origin data. To obtain a higher level of group separation and a better understanding of the variables responsible for classification, supervised OPLS-DA was applied. Afterwards, the parameters for the classification from the software were $R^2Y$ and $Q^2Y$, which were stable and good for fitness and prediction. A sevenfold cross-validation was used to estimate the robustness and predictive ability of our model. A permutation test was performed to further validate the model. The low values of the $Q^2Y$ intercept indicate the robustness of the models, showing a low risk of overfitting and reliability.

**Analysis of GAPDH modification by 4-OI**. RAW264.7 cells were treated with 4-OI (500 μM) for 4 h, and the cells were then lysed in lysis buffer. Cells lysis was immunoprecipitated by anti-GAPDH antibody and Protein A/G Magnetic Beads. After immunoprecipitation, bound GAPDH was eluted off the beads using PBST (1× PBS, 0.5% Triton X-100, pH 7.4). Each bead sample was dissolved in 1× SDS sample buffer and separated by SDS–PAGE. The SDS–PAGE-separated gel was detected by Coomassie blue, and the corresponding bands were excised and in-gel digested by trypsin.

Gel pieces were destained in 50 mM $NH_4HCO_3$ buffer until clear and then dehydrated with 100 μL of 100% acetonitrile for 5 min. After that, the gel pieces were rehydrated in 10 mM dithiothreitol and incubated at 56 °C for 60 min. The gel pieces were again dehydrated in 100% acetonitrile. Liquid was removed, and the gel pieces were rehydrated with 55 mM iodoacetamide. The samples were incubated at room temperature in the dark for 45 min. The gel pieces were washed with 50 mM $NH_4HCO_3$ and then dehydrated with 100% acetonitrile. The gel pieces were rehydrated with 10 ng/μL trypsin and resuspended in 50 mM $NH_4HCO_3$ on ice for 1 h. Excess liquid was removed, and the gel pieces were digested with trypsin at 37 °C overnight. Peptides were extracted with 50% acetonitrile/5% formic acid, followed by 100% acetonitrile. The peptides were dried to completion and resuspended in 2% acetonitrile/0.1% formic acid.

The tryptic peptides were dissolved in 0.1% formic acid (solvent A) and directly loaded onto a homemade reversed-phase analytical column (15-cm length, 75 μm i.d.). The gradient comprised an increase from 6% to 23% solvent B (0.1% formic acid in 98% acetonitrile) over 16 min, an increase from 23% to 35% over 8 min and an increase to 80% over 3 min, and it was then held at 80% for the last 3 min, all at a constant flow rate of 400 μL/min on an EASY-nLC 1000 UPLC system. The peptides were subjected to NSI source followed by MS/MS in Q Exactive™ Plus (Thermo) coupled online to the UPLC. The electrospray voltage applied was

2.0 kV. The $m/z$ scan range was 350–1800 for the full scan, and intact peptides were detected in the Orbitrap at a resolution of 70,000. Peptides were then selected for MS/MS using the NCE setting as 28, and the fragments were detected in the Orbitrap at a resolution of 17,500. A data-dependent procedure that alternated between one MS scan followed by 20 MS/MS scans with 15.0 s dynamic exclusion was used. The automatic gain control was set at 5E4.

The resulting MS/MS data were processed using Proteome Discoverer 1.3. Tandem mass spectra were searched against the UniProt database. Trypsin/P was specified as a cleavage enzyme allowing up to two missing cleavages. The mass error was set to 10 ppm for precursor ions and 0.02 Da for fragment ions. The peptide confidence was set high, and the peptide ion score was set at >20.

**Glycolytic flux analysis with $U^{13}C$-glucose tracing**. RAW264.7 cells were seeded into 10-cm plates and treated with 125 μM 4-OI or vehicle and stimulated with 1 μg/mL LPS. After 6 h of incubation at 37 °C, the cells were washed with PBS. Fresh glucose-free DMEM containing 10% FBS, 2 mM glutamine and 1% peni-cillin/streptomycin was then added, along with fresh 4-OI (125 μM) or vehicle and LPS (1 μg/mL). After overnight glucose starvation, the cells were added with 12 mM $U^{13}C$-glucose (Cambridge Isotope Laboratories, catalogue # CLM-1396) and incubated for six additional hours. The cell plates were then washed twice with PBS, and the cells were collected and stored at −80 °C until analysis.

Cells were homogenized with 0.6 mL of cold (−20 °C) 50% methanol and water containing 100 μM norvaline as an internal standard and ground with a tissue-grinding machine. Then 0.4 mL of chloroform was added to the samples, and the samples were vortexed and centrifuged at $18,630 \times g$ for 10 min at 4 °C. The methanol fractions were dried by centrifugal evaporation and stored at −80 °C before analysis.

The dried samples were derivatized by the addition of 70 μL of methoxyamine hydrochloride in pyridine and incubated for 20 min at 80 °C. After cooling, 30 μL of TBDMS (Sigma) was added to each sample at 80 °C for 60 min before centrifugation at $18,630 \times g$ for 5 min (4 °C). The supernatant was transferred to an autosampler vial for gas chromatography (GC)/MS analysis. A 1 μL aliquot of each derivatized sample was injected into an Rxi-5MS capillary column (20 m × 0.25 mm × 0.25 μm) in a Shimadzu QP-2010 Ultra GC-MS system. The GC oven temperature started at 110 °C for 4 min, increased to 230 °C at 3 °C/min and then to 280 °C at 20 °C/min and was then held at this temperature for 2 min. The mass spectrometer was scanned (50–800 $m/z$) in full-scan mode with helium as the carrier gas.

To determine $^{13}C$ labelling, the known fragments of metabolites were extracted from the appropriate chromatographic peak. These fragments contained the whole carbon skeleton of the metabolite, lacked the alpha carboxyl carbon or contained only the backbone minus the side-chain (for some amino acids)[52]. For each fragment, the retrieved data comprised mass intensities for the lightest isotopomer (without any heavy isotopes, M0) and isotopomers with increasing unit mass (up to M6) relative to M0. These mass distributions were normalized and corrected to the natural abundance of heavy isotopes of H, N, O, Si and C using matrix-based probabilistic methods.[53]

**Cytokine measurements**. IL-1β in the culture medium was measured by com-mercially available enzyme-linked immunosorbent assay (ELISA) kits (Novus, VAL601) according to the manufacturer's instructions. IL-1β and IL-6 levels in the serum were measured according to the manufacturer's instructions using com-mercially available ELISA kits (MEIMIAN, MM-0040M2, MM-0163M1).

**Lactate and GAPDH enzyme activity assay**. Lactate was tested by a lactic acid assay kit from Nanjing Jiancheng Bioengineering Institute (A019-2) according to the manufacturer's instructions. GAPDH enzyme activity was measured with a GAPDH colorimetry assay kit from GenMed Scientifics Inc.

**Seahorse extracellular flux analyser assays for ECAR and OCR**. Briefly, approximately 20,000 cells were seeded into Seahorse XFe96 cell culture micro-plates in 80 μL of growth medium. Before treatment, the medium was removed, and fresh growth medium was added with/without 1 μg/mL LPS for RAW264.7 cells or 100 ng/mL for BMDMs and the indicated concentrations of 4-OI. After 24 h, the medium was changed to glucose-free Seahorse XFe assay medium con-taining 2 mM glutamine for the ECAR assay or 2 mM glutamine, 1 mM sodium pyruvate and 10 mM glucose for the OCR assay (the pH was adjusted to 7.4), and the cells were kept at 37 °C in a $CO_2$-free incubator for an additional 45 min to 1 h prior to the assay. ECAR and OCR were monitored using the Seahorse XFe96 Analyser (Agilent).

**Quantitative real-time PCR (qRT-PCR)**. Total RNA was extracted from cells by the EASYspin Plus Tissue/Cell RNA Extraction Kit (Aidlab Biotechnologies Co., Ltd., Beijing, China) according to the manufacturer's instructions. Purified RNA (1 μg) was reversely transcribed into complementary DNA using HiScript Q RT SuperMix for qPCR (VazymeBiotech Co., Ltd., Suzhou, China). qRT-PCR was performed on a LightCycler 480 Detector (Roche, Mannheim, Germany), and β-actin was selected as a housekeeping gene. The following primer sequences were used: β-actin: Forward primer TGATGGTGGGAATGGGTCAG, Reverse primer

GGTGTGGTGCCAGATCTTCT; IL-1β: Forward primer GCAACTGTTCCT-GAACTCAACT, Reverse primer ATCTTTTGGGGTCCGTCAACT; iNOS: Forward primer CCACAATAGTACAATACTACTTGG, Reverse primer ACGAGGTGTTCAGCGTGCTCCACG.

**Western blot analysis**. Cells were lysed in lysis buffer and then heat-denatured in 2× Laemmli sample buffer. Sample buffers were separated by SDS-PAGE and transferred to polyvinylidene difluoride membranes. The membranes were blocked with 5% skim milk and then incubated with primary antibodies, followed by incubation with anti-rabbit or anti-mouse horseradish peroxidase-conjugated secondary antibodies. The bands were visualized by enhanced chemiluminescence on the Image Lab software. Full-length blots are provided in the Source Data file.

**Statistical analysis**. Quantitative results were expressed as the mean ± SEM from at least three independent experiments. Statistical analyses between groups were performed by GraphPad Prism version 6.0. Student's $t$ test, one-way analysis of variance (ANOVA) or two-way ANOVA was used to compare the differences in each parameter. Survival curves were analysed by log-rank (Mantel–Cox) test. $p$ Values < 0.05 were considered statistically significant.

**Reporting summary**. Further information on research design is available in the Nature Research Reporting Summary linked to this article.

## Data availability
Metabolomics data have been deposited in the EMBL-EBI MetaboLights database under the accession code MTBLS1140. All other data are included in the manuscript are available from the corresponding author on reasonable request. The source data underlying Figs. 2–4 and 5a–d and Supplementary Figs. 2, 3c, d and 4–6 are provided as a Source Data file.

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

## Acknowledgements
This work was funded by the Key Project of National Natural Science Foundation of China (No. 81430092), the National Natural Science Foundation of China (No. 81773857), the "Double First-Class" University Project (CPU2018GF03), the 111 Project from Ministry of Education of China and the State Administration of Foreign Export Affairs of China (No. B18056), and Postgraduate Research & Practice Innovation Program of Jiangsu Province (No. KYCX18-0825).

## Author contributions
S.-T.L. designed and carried out all experiments. C.H. performed Seahorse experiments. L.-Y.K., J.-S.W. and C.H. designed and supervised research. S.-T.L., C.H., D.-Q.X., X.-W. F., J.-S.W. and L.-Y.K. interpreted the data. S.-T.L., C.H., J.-S.W. and L.-Y.K. wrote the manuscript.

## Competing interests
The authors declare no competing interests.
