## [Peer Review File · Nature Communications]

Reviewers' comments:

Reviewer #1 (Remarks to the Author):

In this paper, evidence is provided for itaconate inhibiting GAPDH by modifying it on Cys22. This is shown to be important for the anti-inflammatory effect of itaconate. This is an interesting finding which will have broad interest in the area of metabolic reprogramming in immunity. Overall, the conclusions are well supported by the data. I have however some issues that need to be addressed.

1. The authors need to examine GAPDH in IRG-1-deficient macrophages. These should show an enhancement in GAPDH activity and glycolysis. This is an important experiment as it will confirm an endogenous role for the targeting of GAPDH by itaconate.

2. GAPDH has recently been shown to control translation of TNF in LPS-treated macrophages (Galvan-Pena, S et al Nat Commun. 2019 Jan 18;10(1):338. doi: 10.1038/s41467-018-08187-6). Heptilidic acid has been shown to limit this and so one would expect 4-OI to act similarly - this should be tested here for completeness and to further support the findings.

3. The authors need to cite the Mills et al paper more thoroughly - that paper demonstrated that 4-OI protects against lethality and identified the modification of proteins by itaconate as dicarboxypropylation. These aspects should be discussed.

Minor points

Results section: line 5: Itaconate doesn't modify Nrf2 - it modifies Keap1 - this should be corrected and the Mills paper should be cited. Also in that paragraph reference 25 should be reference 28?

Reviewer #2 (Remarks to the Author):

Authors of the manuscript show significant amount of work devoted to effect of octyl-itaconate treatment of macrophages.

The work on octyl itaconate is interesting but it is critical that authors understand that octyl-itaconate and itaconate is not the same and it is inappropriate to show the data for octyl-itaconate and put itaconate in the title.

It is evident from authors own data - they show here that octylitaconate modifies gapdh as OI, not as itaconate. It is also evident from publication of Mills et al (2018), who erroneously put itaconate in their title while they only showed that Keap1 is binding to OI, not itaconate, as well as anti-inflammatory effects are exerted by octylitaconate, not itaconate.

Therefore, it is absolutely critical that authors either change their title to say that "Octyl-Itaconate inhibits aerobic glycolysis by targeting..." or demonstrate explicitly that effects of octyl-itaconate that they report are mediated through the actual free itaconate, not its derivative.

Among other things, it is critical that authors show the amounts of itaconate accumulated upon addition of OI to macrophages with and without LPS stimulation.

However, in present form i don't think that the paper is in acceptable form.

Reviewers' comments:

Reviewer #1 (Remarks to the Author):

In this paper, evidence is provided for itaconate inhibiting GAPDH by modifying it on Cys22. This is shown to be important for the anti-inflammatory effect of itaconate. This is an interesting finding which will have broad interest in the area of metabolic reprogramming in immunity. Overall, the conclusions are well supported by the data. I have however some issues that need to be addressed.

Response: Thank you very much for your positive comments and constructive suggestions, which are really helpful for the improvement of this manuscript.

1. The authors need to examine GAPDH in IRG-1-deficient macrophages. These should show an enhancement in GAPDH activity and glycolysis. This is an important experiment as it will confirm an endogenous role for the targeting of GAPDH by itaconate.

Response: Thanks a lot for your valuable suggestion and the following experiments were conducted as suggested. We have bought *Irg1* deficient (*Irg1*^{-/-}) mice for this experiment, where *Irg1*^{-/-} bone marrow-derived macrophages (BMDMs) were isolated. The activity of GAPDH, and levels of lactate and ECAR in WT and *Irg1*^{-/-} BMDMs were measured. GAPDH activity was significantly enhanced in *Irg1*^{-/-} BMDMs as compared with WT BMDMs after LPS stimulation for 24 h (Supplementary Fig. 4A), which together with the significant increased levels of lactate and ECAR in LPS induced *Irg1*^{-/-} BMDMs (Supplementary Fig. 4B and C), demonstrating an obviously augmented glycolysis in *Irg1*^{-/-} BMDMs. As a consequence, the level of IL-1 β was significantly increased in LPS induced *Irg1*^{-/-} BMDMs. These results provided convincing evidences that the inhibition of endogenous itaconate production increased GAPDH activity and glycolysis, and promote inflammation. Thanks again for this precious advice.

Supplementary Fig. 4 *Irg1* deficiency leads to enhanced GAPDH activity and glycolysis. WT BMDMs and *Irg1*^{-/-} BMDMs were detected for GAPDH enzyme activity (A) after 100 ng/mL LPS stimulation for 24 h. Levels of lactate (B), ECAR (C) and IL-1β (D) were determined in WT and *Irg1*^{-/-} BMDMs with or without LPS stimulation for 24 h. All data shown are summarized from three independent experiments. Values represent the mean ± SEM at each time point. *p* values were calculated using two-tailed Student's t-test or one-way ANOVA with Sidak's correction for multiple comparisons. Source data are provided as a Source Data file.

2. GAPDH has recently been shown to control translation of TNF in LPS-treated macrophages (Galvan-Pena, S et al Nat Commun. 2019 Jan 18;10(1):338. doi: 10.1038/s41467-018-08187-6). Heptilidic acid has been shown to limit this and so one would expect 4-OI to act similarly - this should be tested here for completeness and to further support the findings.

Response: Thank you for pointing out this important reference¹, where heptilidic acid was reported with significant inhibition on the translation of TNF-α but not its transcription. Our

results showed that the performance of OI was indeed similar to that of heptilidic acid: inhibit TNF- α translation and has no effect on its mRNA levels (Fig. 4, D and E).

effects of OI in macrophages. (A), Heptilidic acid (3 μ M), a GAPDH inhibitor, significantly inhibited IL-1 β release in LPS-stimulated RAW264.7 macrophages. (B) and (C), Treatment with heptilidic acid exerted similar effects of OI on iNOS expression in LPS-stimulated RAW264.7 macrophages and BMDMs. (D) and (E), TNF- α mRNA expression and protein release were measured in 1 μ g/mL LPS-treated (6 h) RAW264.7 cells after treatment with heptilidic acid or OI. (F) and (G), LPS-induced NF- κ B (24 h) expression in the nucleus (F) or cytosol (G) after treatment with vehicle or OI. (H),

Wild-type GAPDH (GAPDH-WT) or the Cys-22 mutant (GAPDH-C22A) was overexpressed in RAW264.7 macrophages for 24 h and then treated with LPS and OI for 24 h, followed by measurement of IL-1 β secretion by ELISA. Representative results are shown from three independent experiments. Data represent the mean \pm SEM. *p* values were determined by two-tailed Student's t-test or one-way ANOVA with Sidak's correction for multiple comparisons test. Source data are provided as a Source Data file.

3. The authors need to cite the Mills et al paper more thoroughly - that paper demonstrated that 4-OI protects against lethality and identified the modification of proteins by itaconate as dicarboxypropylation. These aspects should be discussed.

Response: Thank you for your suggestion. We have now discussed these points in the discussion (lines 180-192), we have rewritten it as follow:

Lines 180-192: OI could covalently modify KEAP1 by dicarboxypropylation². This modification increased nucleus Nrf2 level, and facilitated the expression of downstream target genes with anti-inflammatory and anti-oxidant capacities. KEAP1 normally forms complex with Nrf2 and promotes its degradation. Alkylation of crucial KEAP1 cysteine residue by OI leads to the accumulation of newly synthesized Nrf2, which migrate to the nucleus and activate a transcriptional anti-oxidant and anti-inflammatory program. Nrf2 activation is thus essential for the anti-inflammatory effect of OI. In our study, OI could modify the cysteine 22 residue of GAPDH by similar dicarboxypropylation. The decrease in IL-1 β release induced by OI treatment was successfully and significantly attenuated by overexpression of WT GAPDH but not C22A GAPDH in RAW264.7 macrophages, which demonstrated the essential role of cysteine 22 in GAPDH function. The anti-inflammatory effect of OI is associated with the inhibited glycolysis, which provides prerequisite energy and biosynthetic raw material for M1 macrophages, helping their proliferation and biosynthesis.

Minor points

Results section: line 5: Itaconate doesn't modify Nrf2 - it modifies Keap1 - this should be corrected and the Mills paper should be cited. Also in that paragraph reference 25 should be

reference 28?

Response: Sorry for this typos-error and rectified.

Reviewer #2 (Remarks to the Author):

1. Authors of the manuscript show significant amount of work devoted to effect of octyl-itaconate treatment of macrophages.

The work on octyl itaconate is interesting but it is critical that authors understand that octyl-itaconate and itaconate is not the same and it is inappropriate to show the data for octyl-itaconate and put itaconate in the title.

It is evident from authors own data - they show here that octylitaconate modifies gapdh as OI, not as itacoante. It is also evident from publication of Mills et al (2018), who erroneously put itaconate in their title while they only showed that Keap1 is binding to OI, not itaconate, as well as anti-inflammatory effects are exerted by octylitaconate, not itacoante.

Response: Thank you very much for directing us to the difference between itaconate and OI. Itaconate is a carboxylic acid of high polarity, and thus is not easy for its transportation into the cells through cell membranes composed of low polarity lipids. Therefore, direct administration of itaconate to the cells would not effectively mimic its increase in vivo, and realization of which calls for itaconate surrogates that keep its functional groups intact and its polarity low to the largest possible. Dimethyl itaconate (DMI) and 4-octyl itaconate (OI), membrane-permeable itaconate derivative, were two common such surrogates. Of them, the carboxyl group of the α,β -unsaturated ketone moiety of itaconate is esterified by DI: lacking a negative charge on the conjugated ester group would increase Michael addition reactivity to an extent greater than itaconate itself. In addition, in an experiment using [^{13}C]itaconate and dimethyl [^{13}C]itaconate (DMI) to probe itaconate metabolism: [^{13}C]DMI is not metabolized to itaconate and [^{13}C]Itaconate in the cell culture medium doesn't lead to elevated intracellular uptake of itaconate³. In OI, the α,β -unsaturated ketone moiety of itaconate is kept intact and only the carboxyl irrelevant to Michael addition reactivity was esterified by octyl, making it a suitable itaconate surrogate to study its biological function. For strictness, OI can't completely represent itaconate, so we change the title as "4-octyl itaconate inhibits aerobic glycolysis by targeting GAPDH to exert anti-inflammatory effects".

2. Therefore, it is absolutely critical that authors either change their title to say that "Octyl-Itaconate inhibits aerobic glycolysis by targeting..." or demonstrate explicitly that effects of octyl-itaconate that they report are mediated through the actual free itaconate, not its derivative.

Response: We are grateful to the reviewer for this point and have performed some experiments.

Immuno-responsive gene 1 (*Irg1*) as the gene coding for an enzyme producing itaconic acid by the decarboxylation of cis-aconitate⁴. In this revision, we have bought *Irg1* deficient (*Irg1*^{-/-}) mice, where *Irg1*^{-/-} bone marrow-derived macrophages (BMDMs) were isolated. The activity of GAPDH, and levels of lactate and ECAR in WT and *Irg1*^{-/-} BMDMs were measured. GAPDH activity was significantly enhanced in *Irg1*^{-/-} BMDMs as compared with WT BMDMs after LPS stimulation for 24 h (Supplementary Fig. 4A), which together with the significant increased levels of lactate and ECAR in LPS induced *Irg1*^{-/-} BMDMs (Supplementary Fig. 4B and C), demonstrating an obviously augmented glycolysis in *Irg1*^{-/-} BMDMs. As a consequence, the level of IL-1 β was significantly increased in LPS induced *Irg1*^{-/-} BMDMs. These results provided convincing evidences that the inhibition of endogenous itaconate production increased GAPDH activity and glycolysis, and promote inflammation.

However, considering our holistic study and your suggestion, so we change the title as "4-Octyl-itaconate inhibits aerobic glycolysis by targeting GAPDH to exert anti-inflammatory effects".

Supplementary Fig. 4 *Irg1* deficiency leads to enhanced GAPDH activity and glycolysis. WT BMDMs and *Irg1*^{-/-} BMDMs were detected for GAPDH enzyme activity (A) after 100 ng/mL LPS stimulation for 24 h. Levels of lactate (B), ECAR (C) and IL-1 β (D) were determined in WT and *Irg1*^{-/-} BMDMs with or without LPS stimulation for 24 h. All data shown are summarized from three independent experiments. Values represent the mean \pm SEM at each time point. *p* values were calculated using two-tailed Student's t-test or one-way ANOVA with Sidak's correction for multiple comparisons. Source data are provided as a Source Data file.

3. Among other things, it is critical that authors show the amounts of itaconate accumulated upon addition of OI to macrophages with and without LPS stimulation.

Response: Thank you for this precious advice. We measured the amounts of itaconate accumulation upon addition of OI to macrophages with and without LPS stimulation during

this revision (Supplementary Fig. 2). Macrophages with LPS stimulation have significant and dose-dependent increase of itaconate upon addition of OI, showing OI was hydrolysed to itaconate to a certain extent in LPS stimulated RAW264.7 macrophages. Noteworthy, macrophages without LPS stimulation have no apparent increase of itaconate with the addition of OI. These results are consistent with the publication of Mills et al (2018)² and demonstrated the conversion of OI to itaconate in LPS stimulated RAW264.7 macrophages to act as one sees fit.

Supplementary Fig. 2 OI was hydrolysed to itaconate in LPS stimulated RAW264.7 macrophages. Macrophages were treated with vehicle or OI at the indicated concentrations. After 3 h, cells were stimulated with or without 1 µg/mL LPS for 24 h. Itaconate levels in RAW264.7 macrophages were quantitative by LC-MS/MS. Data represent the mean ± SEM of four independent experiments. *p* values were determined by one-way ANOVA with Sidak's correction for multiple comparisons. Source data are provided as a Source Data file.

References

1. Galvan-Pena, S., *et al.* Malonylation of GAPDH is an inflammatory signal in macrophages. *Nat. Commun.* **10**, 338 (2019).
2. Mills, E. L. *et al.* Itaconate is an anti-inflammatory metabolite that activates Nrf2 via alkylation of KEAP1. *Nature* **556**, 113-117 (2018).
3. ElAzzouny, M., *et al.* Dimethyl itaconate is not metabolized into itaconate intracellularly. *J. Biol. Chem.* **292**, 4766-4769 (2017).
4. Michelucci, A. *et al.* Immune-responsive gene 1 protein links metabolism to immunity by catalyzing itaconic acid production. *Proc. Natl. Acad. Sci. USA.* **110**, 7820-7825 (2013).

REVIEWERS' COMMENTS:

Reviewer #1 (Remarks to the Author):

The authors have addressed by concerns satisfactorily and I'm happy to recommend acceptance.

Reviewer #2 (Remarks to the Author):

All my comments have been addressed

REVIEWERS' COMMENTS:

Reviewer #1 (Remarks to the Author):

The authors have addressed by concerns satisfactorily and I'm happy to recommend acceptance.

Response: Thank you very much for your positive comments.

Reviewer #2 (Remarks to the Author):

All my comments have been addressed.

Response: Thanks a lot.